# Pertussis Prevalence in Adult Population in Greece: A Seroprevalence Nationwide Study

**DOI:** 10.3390/vaccines10091511

**Published:** 2022-09-09

**Authors:** Dimitrios Papagiannis, Eleftherios Thireos, Anargiros Mariolis, Antonios Katsioulis, Konstantina Gartzonika, Niki Malliaraki, Christos Agnantis, Konstantinos Tsaras, Foteini Malli, Erasmia C. Rouka, Ioanna Tsiaousi, Emmanouil K. Symvoulakis, Georgios Rachiotis, Konstantinos I. Gourgoulianis

**Affiliations:** 1Public Health & Vaccines Laboratory, Faculty of Nursing, School of Health Sciences, University of Thessaly, 41110 Larissa, Greece; 2National Health System of Greece, Primary Health Center of Vari Athens, 16672 Attica, Greece; 3National Health System of Greece, Primary Health Center of Areopolis, 23062 Areopoli, Greece; 4Microbiology Department, Faculty of Medicine, School of Health Sciences, University of Ioannina, 45110 Ioannina, Greece; 5Laboratory of Clinical Chemistry-Biochemistry, University Hospital of Heraklion, 71500 Heraklion, Greece; 6Respiratory Disorders Laboratory, Faculty of Nursing, University of Thessaly, 41110 Larissa, Greece; 7Private Primary Health Sector, Queen Sophia Avenue 123, 11521 Athens, Greece; 8Department of Social Medicine, Faculty of Medicine, University of Crete, 71003 Heraklion, Greece; 9Department of Hygiene and Epidemiology, Medical Faculty, School of Health Science, University of Thessaly, 42200 Larissa, Greece; 10Department of Respiratory Medicine, Faculty of Medicine, School of Health Sciences, University of Thessaly, Biopolis, 41110 Larissa, Greece

**Keywords:** *Bordetella pertussis*, antibodies, seroprevalence, adults, vaccination

## Abstract

The reported cases of pertussis vary considerably globally. In the present nationwide study, we aimed to record the Bordetella pertussis prevalence in Greece by measuring serum IgG specific antibody levels to pertussis toxin (anti-PT IgG). General practitioners and laboratories participated in this study from 12 regions of Greece. A geographically stratified sampling plan based on regional units (NUTS level 2) was applied in order to produce a representative sample, taking into consideration age group (30–39, 40–49, 50–59, 60–69, 70–79 and 80+) and sex. In total, 1169 subjects participated in the study. The percentage of participants with anti-PT IgG antibodies higher than 50 IU/mL was 3.7%. The levels of anti-PT IgG antibodies of total sample ranged between 1.46 IU/mL to 126.60 IU/mL, with mean 17.74 IU/mL and standard deviation 14.03 U/mL (*p*-value < 0.001). The total seroprevalence of Greek regions for pertussis disease varied significantly among prefectures. The region with the highest seroprevalence was Peloponnese 21.3%, followed by the region of Central Greece 15.3%. The proportion of adults who have pertussis specific antibodies <50 IU/mL has been >90%, suggesting that a large number of adults may be vulnerable to infection of pertussis despite well-established vaccination programs in Greece. Despite the fact that vaccination reduced the number of reported pertussis cases in the last decades in Greece, our seroprevalence study may indicate that the herd immunity level among Greek adults is suboptimal.

## 1. Introduction

Pertussis is a highly infectious disease caused by the *Bordetella pertussis* bacterium. Humans are the only known reservoir of the bacterium, thus the transmission of the disease occurs between people only, and unlike other childhood diseases, the immunity conferred by a first infection is not definitive but declines over time. In Greece, the introduction of whole-cell pertussis vaccine achieved from *Bordetella pertussis* suspensions (1961) guide to a progressive reduction in the incidence of the disease [1]. In 1995, the acellular vaccine was introduced, consisting of inactivated pertussis toxin (PT) in combination with tetanus polio and diphtheria antigens. The acellular vaccine was found to induce a good serological response, with an efficacy around 84% to 92% and fewer side effects compared to the whole-cell vaccine [2,3]. As a result, the incidence of pertussis in Greece further declined in the period 2004–2020. The disease has a low reported impact in our country. The average annual reported incidence in the period 2004–2020 of the pertussis in Greece was lower than the average reported pertussis incidence of European countries (EU/EEA) (8.2 cases/100,000 population for the year 2018) [4,5]. According to the Greek National Public Health Organization (EODY), 89.5% of children aged 6 years—1st grade have received 5 doses of DTaP, while 95.8% of children in kindergartens aged 2–3 years have received 4 doses of DTaP [5]. It is noted, however, that pertussis is a disease that is underdiagnosed due to difficult clinical diagnosis and often unavailable laboratory confirmation. The fact that the pertussis vaccine provides immunity that declines over time, and that a large percentage of cases concern unvaccinated Roma children, indicate that the pertussis prevention remain in Greece and booster doses of pertussis vaccines is necessary. The application of vaccination with Tdap (Tetanus, diphtheria, acellular pertussis) in teens and adults is expected to reverse the increased incidence of pertussis at this age groups [6,7]. It is also worth noting that according to the new Greek National Adult Vaccination Program, adults over 19 to 25 years should be given one booster dose of Tdap preferably, or Tdap-IPV (Tetanus, diphtheria, acellular pertussis, and poliomyelitis) vaccine, regardless of the period mediated by previous Td/T (Tetanus, diphtheria Tetanus) vaccination and then Td or Tdap every 10 years [8]. The national vaccination programs for pertussis are designed to protect vulnerable populations such infants and pregnant women. Overall transmission rates in the household were high. Young infants are a high risk for infection of pertussis, and older studies have shown that parents, siblings, and grandparents are the main source of pertussis transmission in infants [9]. To the best of our knowledge, there are no published data on the seroprevalence of pertussis at a nationwide scale in Greece. These data are of crucial importance in order to inform future vaccination campaigns against pertussis. Consequently, the aim of the present study was to assess the seroprevalence of IgG antibodies against pertussis in the adult’s Greek population. Moreover, correlation of IgG antibodies and sex, age, and geographical area were studied.

## 2. Materials and Methods

A geographically stratified sampling plan based on regional units (Eurostat’s NUTS-2) was applied in order to produce a representative sample, taking into consideration age group (30–39, 40–49, 50–59, 60–69, 70–79, and 80+) and sex. The required minimum sample size was determined to be about 1100 serum samples from the 13 NUTS level 2 regions, and the sample size for each regional unit from the corresponding region was calculated according to the last census population considering the expected prevalence of 50% with a precision of ±3%, a confidence level of 95%, and a power of 80%. Calculated 50% prevalence according to the sparse data for the prevalence of pertussis in Greece. The proportion of 50% leads to a larger sample size of participants and reduces the probability for error about the value of prevalence [10]. The serum samples were collected from a nationwide network of General practitioners (GP), including microbiological laboratories of public Hospitals and primary health care facilities. One thousand and five hundred residents were invited to participate in the study and 1169 accepted the invitation to participate (79.38% response rate). During September 2021 to March 2022, a total of 1169 blood samples were collected from a nationwide network of General practitioners (GP), including microbiological laboratories of private and public hospitals and primary health care facilities.

### 2.1. Sample Collection

The samples were derived from individuals who visited the GP and primary health care facilities for routine screening and reasons unrelated to Bordetella pertussis. Exclusion criteria were age under 30 years old, residence, and symptoms for pertussis illness. We excluded participants under 30 years old from the study. Each prospective participant was informed about the aims and procedures of the study and could then freely choose whether to participate or not. The serum samples were anonymized. Each sample had a unique code and the required data—sex, age, residence, and date of blood sampling—were recorded by the physicians. All participants have given their written informed consent for participation in the survey. Blood samples were centrifuged (10 min at 2500 rpm) and sera were stored at −20 °C. Serum samples were transferred to the Public Health and adult’s immunization laboratory of University of Thessaly and stored at −80 °C, awaiting analyses. The research protocol was approved by the ethical committee of the University of Thessaly (protocol number 49/04.06.2021) and complied with ethical standards described in the Declaration of Helsinki [11].

### 2.2. Statistical Analysis

Quantitative variables are presented as minimum, first quartile, median, third quartile, maximum, and mean with standard deviation. Qualitative variables are presented as frequencies with percentages or 95% CIs (Confidence Intervals) or both. Numerical variables were checked for normality of the distribution. Spearman’s coefficient correlation was used to identify any relationship between quantitative variables. In the univariate analysis, Chi-square test or Fisher’s exact test was used to identify any association between qualitative variables. Mann-Whitney test or Kruskal-Wallis test were used to explore any differences among groups of quantitative variables with non-normally distributed quantitative variables. Additionally, boxplots were used to graphically demonstrate the distribution of numerical data. In the multivariate analysis, multinomial logistic analysis was performed to assess any association between factors and pertussis result. Gender, age or age groups, and administrative region were used as independent variables (factors), and pertussis result or anti-PT IgG as dependent variables. *p* values of less than 0.05 were considered as statistically significant. Statistical analysis was conducted using IBM SPSS Statistics software version 26.0 (IBM Corp., Armonk, NY, USA).

### 2.3. Laboratory Analysis

The enzyme-linked immunosorbent assay (ELISA) is a good additional tool for measuring pertussis antibodies from serum for evaluation of vaccines and to diagnose pertussis [12,13,14]. For the Public Health and adults Immunization Laboratory of the University of Thessaly, we used the SERION ELISA classic Bordetella pertussis Toxin IgG qualitative and quantitative immunoassays for the detection of human antibodies in serum directed against Bordetella pertussis and Bordetella parapertussis for the laboratory confirmation of whooping cough with 99% Sensitivity and Specificity (Institute Virion\Serion GmbH Würzburg, Würzburg, Germany). According to the manufacturer’s classification, values of anti-PT IgG were calculated for the following three intervals: <40 IU/mL, as negative, 40 IU/mL to <50 IU/mL as borderline, and finally >50 IU/mL as positive. We compared the proportions of samples with anti-PT IgG levels in each interval by age group and by region using the chi-squared test or Fisher’s exact test.

## 3. Results

From the total 1169 participants (came from 12 out of 13 regions in Greece), 557 were male (47.6%) and 612 were female (52.4%) from 12 out of 13 regions in Greece (Table 1). The levels of pertussis IgG antibodies of total sample ranged between 1.46 IU/mL to 126.60 IU/mL, with mean 17.74 IU/mL and standard deviation 14.03 IU/mL (Table 1) and (Figure 1).

According to the laboratory method used in our study, we report 44 participants (3.7% of the study population) with positive anti-PT IgG > 50 IU/mL. Furthermore, in analysis of pertussis prevalence by sex, we observed statistically significant difference among participants with males to present a twofold percentage 5% than females 2.6% (*p*-value 0.026). The region of Peloponnese recorded the highest proportion, with 16.4% of antibodies levels more than >50 IU/mL, followed by the region of Thessaly, with 7.4%, and the regions with the lowest proportion were the North Aegean, West Macedonia, and Ionian Islands, with zero samples of anti-PT. IgG > 50 IU/mL was 7.69%, (90/1169) (Table 2) and this may indicate protection against pertussis infection In addition, only one sample was measured for anti-PT IgG > 100 IU/mL, and this probably indicated a pertussis infection or vaccination within the past few years.

In the analysis of antibodies by age group, we observed anti-PT IgG ranged from 1.46 IU/mL to 126.60 IU/mL in the age group of (60–69) and 4.11 IU/mL to 60.39 IU/mL in the participants older than 80 years (Table 3).

The highest mean titers of anti-PT IgG and 18.87 IU/mL were recorded in the age group of 80+ years, followed by the age group of 50–59 years with 18.15 IU/mL, and the lowest mean title of anti-PT IgG was recorded in the age group of 60–69 with 16.86 IU/mL (Figure 2). Further, we analyzed anti-PT IgG (separating in two subgroups (>40 IU/mL vs. <40 IU/mL)) by age groups using the Chi-square test for trend. The result was not statistically significant, as the *p*-value was 0.908 (results not shown).

Anti-PT IgG (mean titers) ranged between regions from 25.18 IU/mL of region of Peloponnese followed by the region of Thessaly with 23.37 IU/mL, and the lowest proportion was recorded by the region of Central Macedonia with 14.22 IU/mL, anti-PT IgG (Figure 3).

In statistical analysis, we implemented regression models to identify the factors which affect the pertussis result. Even though pertussis result is an ordinal variable, ordinal regression was not conducted as the proportional odds assumption was violated, so a multinomial logistic regression was performed. According to the multinomial logistic regression for pertussis, there is no difference with estimated prevalence in males and females to be positive or negative in pertussis (OR = 1.86; 95% CI: 0.98–3.52; *p* = 0.058). The Epirus region is significantly more likely than the Attica region to have anti-PT IgG (OR = 3.38; 95% CI: 1.06–10.82; *p* = 0.040). The Peloponnese region is significantly more likely than Attica region to have anti-PT IgG (OR = 5.54; 95% CI: 2.34–13.10; *p* < 0.001). In addition, Central Greece region is significantly more likely than the Attica region to have boarder anti-PT IgG to negative (OR = 3.01; 95% CI: 1.13–8.01; *p* = 0.027) (Table 4).

The seroprevalence of pertussis varied significantly among regions. The region with the highest seroconversion was the Peloponnese, 21.3%, followed by the region of Central Greece with 15.3%, and the regions with zero seroconversion were the Ionian Islands and West Macedonia (Figure 4).

## 4. Discussion

Pertussis still remains a global public health problem, despite different vaccination schedules worldwide [15]. According to ECDC data, Greece is a low-incidence country in children under the age of 12 months and the general population [16]. This is the first nationwide study in Greece that has aimed to investigate the prevalence of Bordetella pertussis in the general adult population. The results from the present study indicate low prevalence of anti-PT IgG (>50 IU/mL) in the adult population. Reported pertussis prevalence rates varied considerably, ranging from 0% to region of Ionian Islands and West Macedonia, with the highest rates, 16.4%, generally reported in the region of Peloponnese. Results reported from Portugal suggest that seroprevalence for pertussis was 5.8% during 2010–2013, while incidence varied from 0.1–2.2 per 100,000 citizens. The notification rates are high in infants and children. In contrast, there is significant underreporting of adult B. pertussis [17]. We observed that the region of Peloponnese recorded the highest proportion of anti-PT IgG 16.4%, more than >50 IU/mL, followed by the region of Thessaly with 7.4%. The regions with the lowest proportion were the North Aegean, West Macedonia, and Ionian Islands, with zero samples with anti-PT IgG more than >50 IU/mL. Variations in anti-PT IgG titers could correlate not only with the immune status of the population vaccinated but also with the number of cases of pertussis infection. A study that was conducted in five Italian regions reported that, in serum samples collected in the region of Tuscany, 5% of the general population and 7.1% of participants aged 20–29 years had antibody titers compatible with a recent infection (PT-IgG level > 100 IU/mL), indicating an increase in the circulation of B. pertussis in adults in Tuscany region [18]. Similar results were recorded from another older study in six Italian regions. They found that in the age groups 25–44 and >65 years, the seroprevalence for a PT-IgG level > 100 IU/mL was 5.7 and 6.0 %, respectively [19]. In our study, only one sample had a value > 100 IU/mL anti-PT IgG and probably indicated a pertussis infection or vaccination within the past few years.

In the present study, according to the multinomial logistic regression for pertussis result, males are more likely than females to be positive in anti-PT IgG. A recent study by Marchi et al. showed participants aged over 12 years had low IgG titers against pertussis, although with some differences among age groups but not between sex [20]. Similar results were reported by Nokhodian et al., with no significant differences seen between IgG Ab according to sex of residences [21]. Older studies suggest a sex difference for the years 2001–2003, and about 54% of reported cases in USA were among females [22]. Some studies have reported higher seropositivity in male participants [23,24]. However, it is not clear whether this is difference in infection rates or the severity and complication of illness to be reported. In some references morbidity and mortality rates have been higher in females than males [25], but there is no evidence that females are more susceptible to infection than males, and seroepidemiology studies in difference communities have not shown that the prevalence of pertussis antibodies was increased in females in comparison to males.

Despite the publicized literature, anti-PT IgG levels have been correlated with protection against pertussis, and there is no agreement on the level of pertussis antibodies that accords protection against pertussis [26]. Subjects with anti-PT IgG antibodies higher than 40 IU/mL were 7.70% from the total sample (90/1169). A recent study that was conducted in China recorded positive pertussis antibodies in 1.79% in 2018, 2.04% in 2019, and finally 1.66% in 2020, respectively [27]. A cross-sectional retrospective seroprevalence study among middle-aged adults in 18 EU/EEA countries, including Greece, showed that there is circulation of B. pertussis, despite highly implemented childhood vaccination programs [28].

One of the limitations of the present study was the unknown status for the vaccination coverage of the participants. According to the general practitioners that cooperated in the present study, regions such as Peloponnese or Central Greece had high vaccination coverage in the adult population for the vaccines that are suggested by the Greek National Vaccination Program—these include the booster dose vaccine against tetanus, diphtheria, and pertussis (TdaP). The results support our hypothesis that the General Practitioners have an important key role in the implementation of the National Vaccination Program in the adult population. In many cases, the personal relationships which were developed between family Physician and patients are important for better compliance with the doctor’s instructions [29]. An older study that was conducted in Greece among the elderly recorded low vaccination coverage for some booster vaccines such as Tetanus, Diphtheria, and Pertussis. The vaccination coverage for Td vax adult vaccine was 0.30%, for Tdap-IPV it was 0.01%, and for the monovalent vaccine of tetanus it was only 0.30% [30].

The recent COVID-19 pandemic has highlighted and supported the value of vaccination of vulnerable adult populations as a preventive measure against infectious diseases. The International Council on Adult Immunization emphasizes the urgent need for a global adult immunization strategy that covers available vaccines against a variety of pathogens [31]. Adult vaccination strategies against pertussis might be the most effective measure to not only protect the individuals from the disease but also its complications. The increase in the proportion of people without specific anti-PT IgG antibodies also suggests the risk of declined herd immunity. Our results support the WHO position on adult booster vaccination [32]. Pertussis still remains a threat for adults and a booster dose in this age group including maternal immunization is recommended in many countries [33,34]. New recommendations for routine vaccination of adolescents have been added to the new immunization schedule of National Immunization of Greece for pertussis [8]. In the new role of Primary Health care, the family doctors will be having a key role for the implementation of booster doses suggested by the Greek vaccination program for adults. 

According to the WHO, in the recent pandemic, primary health care teams deliver more than ever in all countries. Investing in the prestige of primary health care services will enable health systems to keep up with the growing demands of an ageing population. Enhancing the trust of the population by demonstrating results, especially for vaccination coverage and considering their views, will create popular support for continuing the transformation and increased resourcing of primary health care. In the European Immunization Agenda 2030, primary health care as a first point of contact is best positioned to provide equitable population coverage with essential and evidence-informed immunizations that prevent avoidable infectious diseases. The Regional Office’s work program will focus on primary health care-based strategies to maintain and achieve high vaccination rates, address system barriers, and address vaccine hesitancy [35]. A recent study that was conducted by Poon et al. revealed that family doctors vaccinated against COVID-19 would be more likely to make vaccination recommendations to their patients, and these results support the position of WHO for trust by demonstrating results between patients and health professionals [36].

Study limitations include that seroprevalence from the present sample of participants might not represent the prevalence of pertussis for the whole country, since we were unable to collect data from one out of the 13 Greek regions. In addition, children were not included, and the association of vaccination rates and increases of seroprevalence is uncertain. Pertussis is rather unique, in that the same antibody to pertussis antigens may be measured for diagnostic purposes for performing seroepidemioogical studies and for measuring vaccine responses, as well as for estimating susceptibility to infection. The specific humoral immune responses induced by vaccines cannot be clearly distinguished from those induced after infection. Furthermore, the vaccination status of the participants was not known. Notwithstanding these shortcomings, our study has the advantage of being the first nationwide study of pertussis seroprevalence among Greek adults, and our results could be helpful for policymakers and health care workers in order to inform future vaccination campaigns against pertussis in Greece.

## 5. Conclusions

TdaP vaccines are effective against pertussis in adolescents and adults and several studies showed that protection may wane rapidly by time [37,38]. In the present study, the proportion of adults who did not have pertussis specific antibodies (<50 IU/mL) was estimated at >90% of the study population. This finding may suggest that a large number of adults are vulnerable to infection of pertussis. Lionis et al. proposed organizational harmonization of care between central and regional health authorities in order to multilaterally match health care services with all fields and sectors, which potentially influences both health promotion and disease prevention, remaining a promising real-world expectation to be achieved from an inspired co-action of public health and primary care [39]. Services should be custom-made to the local population’s health care needs, monitoring the available state health resources, and policy scheduling agendas should consider a joint public perception of targeted services in the future [39]. This becomes interesting as suggestions took place before COVID-19 domination and without being able, even today, to estimate how much pandemic narrowed other health promotion initiatives in order to fairly prioritize emergency or rescue responses. Some regions recorded zero specific antibodies against pertussis. Although the intervention measures used such as the vaccination reduced the number of reported pertussis cases in the last decades in Greece, our seroprevalence study indicated that the herd immunity against *Bordetella Pertussis* may be suboptimal. The results of the present study could be helpful for policymakers and health care workers in order to inform future vaccination campaigns against pertussis among adults in Greece.

## Figures and Tables

**Figure 1 vaccines-10-01511-f001:**
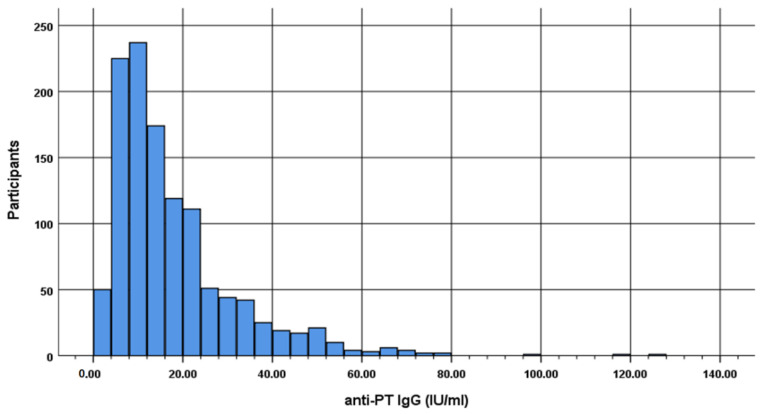
Histogram of anti-PT IgG distribution IU/mL (in groups by 4 IU/mL).

**Figure 2 vaccines-10-01511-f002:**
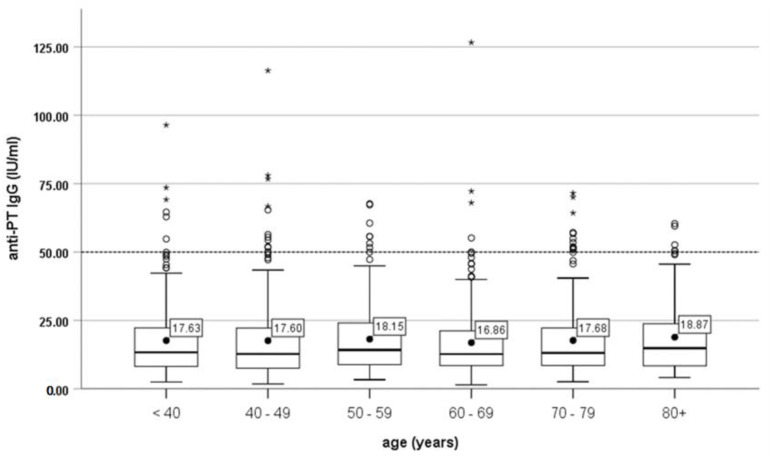
Pertussis anti-PT IgG (mean) by age.

**Figure 3 vaccines-10-01511-f003:**
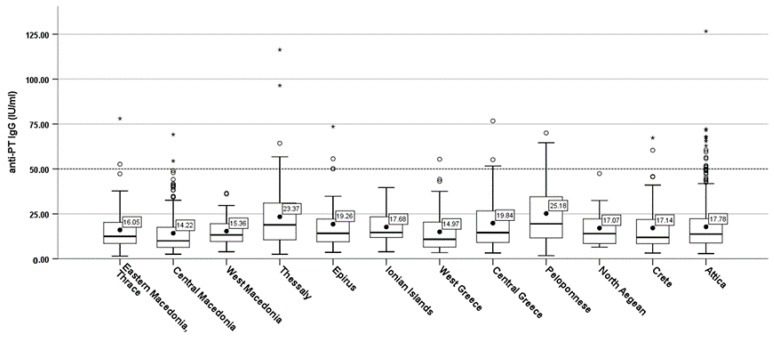
Anti-PT IgG (mean) by region.

**Figure 4 vaccines-10-01511-f004:**
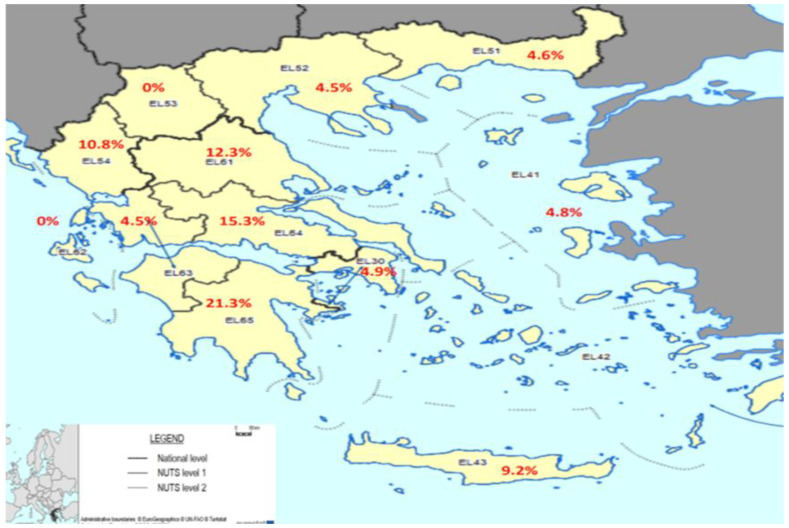
Total seroprevalence of pertussis (positive and borderline) by regions (Eurostat Nuts 2).

**Table 1 vaccines-10-01511-t001:** Participants and sample distribution.

Variable	N	%	Anti-PT IgG (IU/mL)		
			Min/Max	Mean ± SD	*p*-Value *
Sex					0.004 *
Male Female	557 612	47.6 52.4			
Region					<0.001 **
Eastern Macedonia, Thrace	65	5.6	1.46–78.04	16.05 ± 12.65	
Central Macedonia	196	16.8	2.56–69.16	15.36 ± 8.15	
West Macedonia	30	2.6	3.91–36.50	15.36 ± 8.15	
Thessaly	81	6.9	2.49–116.32	23.37 ± 18.84	
Epirus	37	3.2	3.55–73.55	19.26 ± 15.73	
Ionian Islands	21	1.8	3.89–39.62	17.68 ± 10.29	
West Greece	67	5.7	3.40–55.42	14.97 ± 11.12	
Central Greece	59	5.0	3.24–76.72	19.84 ± 15.99	
Peloponnese	61	5.2	1.77–70.04	25.18 ± 17.56	
Attica	466	39.9	2.87–126.60	17.78 ± 13.64	
North Aegean	21	1.8	6.49–47.43	17.07 ± 10.49	
South Aegean	0	0.0	-	-	
Crete	65	5.6	3.20–67.26	17.14 ± 14.12	
Total	1169	100.0			

* Mann-Whitney test. ** Kruskal-Wallis test.

**Table 2 vaccines-10-01511-t002:** Pertussis IgG prevalence by sex and region.

Variable	n/%, *p*-Values							
	Positive >50 IU/mL		Boarderline 40 IU/mL to <50		Negative <40 IU/mL		Total	
Sex Male Female	28	5.0	27	4.8	502	90.1	557	0.026 *
	16	2.6	19	3.1	577	94.3	612	
Eastern Macedonia, Thrace	2	3.1	1	1.5	62	95.4	65	0.001 **
Attica	15	3.2	17	3.6	434	93.1	466	
West Greece	1	1.5	2	3.0	64	95.5	67	
West Macedonia	0	0.0	0	0.0	30	100.0	30	
Ionian Islands	0	0.0	0	0.0	21	100.0	21	
Epirus	4	1.8	0	0.0	33	89.2	37	
Central Macedonia	1	0.5	8	4.1	187	95.	196	
Crete	2	3.1	4	6.2	59	90.8	65	
Peloponnese	10	16.4	3	4.9	48	78.7	61	
Central Greece	3	5.1	6	10.2	50	84.7	59	
Thessaly	6	7.4	4	4.9	71	87.7	81	
North Aegean	0	0.0	1	4.8	20	95.2	21	

* Chi-square test. ** Fisher’s Exact Test.

**Table 3 vaccines-10-01511-t003:** Pertussis anti-PT IgG by age group.

Age	Anti-PT IgG (U/mL)					
	Minimum	Maximum	Mean	Standard Deviation	N	*p*-Value *
<40	2.49	96.42	17.63	14.41	201	0.376
40–49	1.77	116.32	17.60	15.33	231	
50–59	3.32	67.66	18.15	12.96	208	
60–69	1.46	126.60	16.86	14.20	204	
70–79	2.56	71.52	17.68	13.86	187	
80+	4.11	60.39	18.87	12.85	137	

* Kruskal-Wallis test.

**Table 4 vaccines-10-01511-t004:** Multinomial logistic regression (MLR) analysis (positive samples).

Variable	OR	95% CI	*p*-Value
**Sex**			
Male	1.86	0.98–3.52	0.058
Female	ref.		
**Regions**			
Eastern Macedonia, Thrace	0.91	0.2–4.07	0.897
Central Macedonia	0.15	0.02–1.16	0.069
West Macedonia	NA **	NA **	NA **
Thessaly	2.40	0.9–6.42	0.080
Epirus	3.38	1.06–10.82	**0.040**
Ionian Islands	NA **	NA **	NA **
West Greece	0.43	0.06–3.35	0.424
Central Greece	1.70	0.47–6.09	0.416
Peloponnese	5.54	2.34–13.1	**<0.001**
North Aegean	NA **	NA **	NA **
Crete	0.96	0.21–4.29	0.952
Attica	ref. **		

Likelihood Ratio Tests *p*-value (Sex-0.049, Region < 0.001). ** The reference category is negative. NA; not applicable.

## Data Availability

The data that support the findings of this study are available on request from the corresponding author.

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
