# Peer review of "Pertussis Prevalence in Adult Population in Greece: A Seroprevalence Nationwide Study"

_vaccines, 2022, doi:10.3390/vaccines10091511_

Round 1
Reviewer 1 Report
Dimitrios Papagiannis and colleagues submit for publication a study entitled "Pertussis prevalence in adult population in Greece: A seroprevalence Nationwide Study".
It provides serological data obtained with an ELISA test for IgG anti Pertussis perfomed in one central laboratory on blood samples specifically collected over a 6 months period on a sample of 1169 (557 males) adults aged 30 years or older residing in 12/13 Greek prefectures.
The manuscript raises several comments
This study was probably conducted as part of the new vaccination plan for Greece that includes a booster in young adults, as a guide or an evaluation basis for the new preventive actions. This should be better explained in the introduction. The challenge of vaccinating young adults to prevent infection in newborns should also be mentioned in the introduction.
The question of the cut-off to use for identifying participants who can be considered as temporally immune because of an infection or vaccination within the past few years is key in this type of study.
· the Material and method sections defines a negative cut off (> 30 IU/mL) and a positive cut off (> 50 IU/mL) based on the recommendation of the manufacturer of the ELISA test (It is not clear how tests results of 30 and 50 would be classified).
· an additional cut off (> 40 IU/mL) is mentioned in the results section (Line 158 page 5) indicating a pertussis infection or vaccination within the past few years. These cut of should be defined in the material and method section and the respective use of the different cut off should be clarified.
· a cutoff of 30 IU/mL is also used in the literature to identify these cases of infections or vaccination. This question should be discussed in the discussion when comparing findings between studies.
· the discussion also mentions the cut off of 100 UI/mL as interesting for identifying cases of recent infections. This cut off was not used by the authors. Could they explain why.? What is because the study was more concerned with seroprevalence than by estimating incidence of cases of infection. The discussion included reference to incidence in the literature.
The study only included adults aged 30 years or older. The choice of excluding young adults -who are a target population for immunization campaigns - should be explained.
In the context of this seroprevalence study and the reinforcement of immunization actions, efforts should concentrate on analyzing the proportion of participants that can be considered as at least temporarily immune, or non-immune rather than analyzing in too much extend the distribution of negative IgG results. For example table 3 should include a presentation of the proportion of participants who can be considered as immune and non immune by age group.
Confidence intervals should be provided for % estimates in the text and in the tables. According to multivariate analysis, there is no difference with estimated in male and females (P = 0,058 is not < 0.05). This should be corrected. Was age included in the multivariate analysis?
Sample size was computed for an expected prevalence of 50%, that turned out to be much lower. This should also be commented.
The manuscript is overall very long. There are several repetitions. In the methods section, subtitles should also created to introduce the paragraphs where methods for sampling participants and for sampling blood are explained . The results section should also be better organized and the wording should be more efficient. Tables should be made more readable by avoiding repeating % if not necessary. The name of prefecture should be listed alphabetically in Table 2. The discussion does not need to repeat information defined in the material and methods section. The manuscript should best be reviewed by a professional medical writer.
Author Response
On behalf of the authors team we would like to thank the reviewer for reviewing our paper entitled “Pertussis prevalence in adult population in Greece: A seroprevalence Nationwide study” Manuscript ID: vaccines-1836977.
The thoughtful and constructive comments helped us to considerably improve the overall quality of our manuscript. We modified the text accordingly reviewer’s suggestions and we add fine new references (9,10,37,38,39).

Reviewer 2 Report
Thank you for the opportunity to be involved in the revision of this manuscript. The authors performed a nationwide study to assess the Bordetella pertussis prevalence in Greece by measuring serum IgG specific antibody levels to pertussis toxin. The article is of interest to the audience of Vaccines. As major revision, I would ask the authors apply a thorough professional English editing, as several typos make the article difficult to read, just as examples:
Line 54: “guide to a”
Lines 50-60: “The average annual reported impact on in the period 2004-2020 the incidence”
Brackets in lines 63-64
Line 68: “need redefining”
Line 76: “are to assessment”
Lines 87-89
Line 98: “consent” missing
Lack of punctuation in line 151, 154, 155
Line 164: “were ranged”
Inappropriate brackets in “(60-69)” line 165
Lines 184: “wasn’t”
Lines 233: “demonstrated”
Line 237: “appears to be”
Other minor comments:
Lines 186-187: “males are more likely than females to be positive in pertussis to 186 negative (OR=1.86; 95% CI: 0.98-3.52; p=0.058)”. This is not true, p>0.05, please correct.
Author Response

(The authors gave the same response as above.)

Reviewer 3 Report
Even though the field of this article is really important, major revisions must be taken place. Some spelling and syntax errors noticed in your text.
Analytically:
Abstract
Page 1, Line 28- please write Bordetella pertussis in italics.
Introduction
Page 2, Lines 49 and 53 - please write Bordetella pertussis in italics.
Please rephrase next sentence: "The aim of the present seroprevalence study are to assessment the IgG antibodies levels of pertussis in the adult’s Greek population according sex, age and geographical area in six months’ period."
For example: "The aim of the present study was to assess seroprevalence of IgG against pertussis in the adult’s Greek population during the 6 months' period. Moreover, correlation of IgG antibodies and sex, age and geographical area were studied"
Materials and methods
Inclusion and exclusion criteria must have to be included
Results
Please correct names of all administrative regions of Greece in tables 1 and 2 and in the text. For example: Anatoliki Makedonia, Thraki must be corrected to Eastern Macedonia and Thrace.
The quality of all figures is extremely low.
According to Instructions for Authors, file for Figures and Schemes must be provided at a sufficiently high resolution.
In the analysis of antibodies by age group must be provided in two subgroup (> 40 U/ml vs < 40 U/ml) separately.
The result section must be re-written.
As you indicate in this study, the primary aim was to assess seroprevalence of IgG. According to your results, based to your laboratory method, the seroprevalence of IgG was found to be 3.7% (44 participants > 50 IU/mL, male/female).
At the page 7 and line 198 you describe the seroprevalence of IgG in region of Kentriki Ellada (15.3%), but you never describe this region in the text and tables 1 and 2. Perhaps you means Thessaly region?
Please correct all references according to journal stile.
References should be described as follows:
Journal Articles:
1. Author 1; Author 2. Title of the article. Abbreviated Journal Name Year; Volume: page range.
Some references include month. (reference 15, APMIS, 2021, Sep;129(9):556-565.) as well as 18, 20, 24-28. Please delete months from all references.
Conclusions are too short.
Author Response

(The authors gave the same response as above.)

Reviewer 4 Report
Papagiannis and coauthors reported the Bordetella pertussis prevalence in Greece by measuring serum IgG specific antibody levels to pertussis toxin (anti-PT IgG). Geographically stratified sampling plan based on regional units (NUTS level 3) was applied in order to produce a representative sample, taking in to consideration age group and sex. The study revealed that the proportion of adults who did not have pertussis specific antibodies has been increasing, even up to 90% since the suggesting that a large number of adults are vulnerable to infection of pertussis. It also indicated that the herd immunity still remains poor against Bordetella Pertussis. This research is meaningful and informative. Therefore, I recommend the paper be accepted by Vaccines after addressing the following issues in a minor revision.
1. There are many grammatical mistakes throughout the paper, the authors should double check it. For example, on page 1, line 32-33: “1169 resident’s males and females and participated in the study.” The “and” should be deleted. On page 1, line 33-34: “The percentage of participants with anti-PT IgG antibodies higher than 40 IU/mL” should be “The percentage of participants with anti-PT IgG antibodies is higher than 40 IU/mL”
2. What does the x-axis of Figure 1 stand for? The authors should explain it in the caption.
3. Could the authors explain what are the p-values of other age groups (older than 40) in Table 3?
4. The authors should double check the format of the references. For example, ref. 31.
Author Response

(The authors gave the same response as above.)

Round 2
Reviewer 2 Report
Thank you for involving me in the revision of this manuscript. I have no further comments as my previous ones were adequately addressed.
Author Response
We would like to thank the reviewer once again for the constructive comments who help us to improve the total quality of manuscript.

Reviewer 3 Report
I have carefully read an attachment file from (https://ec.europa.eu/eurostat/documents/3859598/9397402/KS-GQ-18-007-EN-N.pdf/68c4a909-
30b0-4a90-8851-eddc400a5faf?t=1543835542000 ).
The name of all Greek regions are presented in correct form, not in Greeklish.
I send you as an attachment file classification of all Greek regions according to Nuts 2.
I strongly recommend to authors to use official names of regions in manuscript (both in text and on the tables)
In adition, authors correct some refereces according to journal stile, but sume references were not corrected.
Ref 7; "Pediatr Infect Dis J. 2007 Sep;26(9):806"
Ref 17; " APMIS, 2021, Sep;129(9):556-565."
Ref 20; " BMJ Open, 2019, Oct 30;9"
Ref 21; "J Res Med Sci, 2021 Mar 31; 26:21"
Ref 26; " J Infect Dis, 2000, Mar;181(3):1010-3"
Ref 27; " Vaccines (Basel), 2022 May 29;10(6):872"
Ref 28; " Nat Commun, 2021, May 17;12(1):2871"
Ref 29; " Med, 2010, Nov-Dec;8(6):507-10"
Ref 30, Ref 31
Please correct.
Moreover, reference number 40 is without text.
My other corrections and recommendations were added.
Author Response

(The authors gave the same response as above.)
